# Targeting Brain Tumors with Mesenchymal Stem Cells in the Experimental Model of the Orthotopic Glioblastoma in Rats

**DOI:** 10.3390/biomedicines9111592

**Published:** 2021-11-01

**Authors:** Natalia Yudintceva, Ekaterina Lomert, Natalia Mikhailova, Elena Tolkunova, Nikol Agadzhanian, Konstantin Samochernych, Gabriele Multhoff, Grigoriy Timin, Vyacheslav Ryzhov, Vladimir Deriglazov, Anton Mazur, Maxim Shevtsov

**Affiliations:** 1Institute of Cytology of the Russian Academy of Sciences (RAS), 194064 St. Petersburg, Russia; e.lomert@gmail.com (E.L.); natmik@mail.ru (N.M.); entolk62@mail.ru (E.T.); nicoleagadzhanyan@gmail.com (N.A.); grigorii.timin@unige.ch (G.T.); shevtsov-max@mail.ru (M.S.); 2Personalized Medicine Centre, Almazov National Medical Research Centre, Polenov Russian Scientific, Research Institute of Neurosurgery, 197341 St. Petersburg, Russia; neurobaby12@gmail.com; 3Department of Chemistry and Molecular Biology, ITMO University, 197101 St. Petersburg, Russia; 4Central Institute for Translational Cancer Research (TranslaTUM), Department of Radiation Oncology, Klinikum Rechts der Isar, Technical University of Munich, 81675 Munich, Germany; gabriele.multhoff@tum.de; 5Department of Genetics and Evolution, University of Geneva, 1205 Geneva, Switzerland; 6Petersburg Nuclear Physics Institute Named by B.P.Konstantinov of National Research Center “Kurchatov Institute”, 188300 Gatchina, Russia; ryzhov_va@pnpi.nrcki.ru (V.R.); deriglazov_vv@pnpi.nrcki.ru (V.D.); 7Magnetic Resonance Research Center, Saint-Petersburg State University, 199034 St. Petersburg, Russia; a.mazur@spbu.ru

**Keywords:** mesenchymal stem cells, biodistribution, nonlinear magnetic response, superparamagnetic iron oxide nanoparticles, multiforme glioblastoma, C6 glioma, magnetic resonance imaging, targeted drug delivery

## Abstract

Despite multimodal approaches for the treatment of multiforme glioblastoma (GBM) advances in outcome have been very modest indicating the necessity of novel diagnostic and therapeutic strategies. Currently, mesenchymal stem cells (MSCs) represent a promising platform for cell-based cancer therapies because of their tumor-tropism, low immunogenicity, easy accessibility, isolation procedure, and culturing. In the present study, we assessed the tumor-tropism and biodistribution of the superparamagnetic iron oxide nanoparticle (SPION)-labeled MSCs in the orthotopic model of C6 glioblastoma in Wistar rats. As shown in in vitro studies employing confocal microscopy, high-content quantitative image cytometer, and xCelligence system MSCs exhibit a high migratory capacity towards C6 glioblastoma cells. Intravenous administration of SPION-labeled MSCs in vivo resulted in intratumoral accumulation of the tagged cells in the tumor tissues that in turn significantly enhanced the contrast of the tumor when high-field magnetic resonance imaging was performed. Subsequent biodistribution studies employing highly sensitive nonlinear magnetic response measurements (*NLR-M*_2_) supported by histological analysis confirm the retention of MSCs in the glioblastoma. In conclusion, MSCs due to their tumor-tropism could be employed as a drug-delivery platform for future theranostic approaches.

## 1. Introduction

Despite multimodal interdisciplinary treatment approaches [1,2], malignant gliomas, particularly multiforme glioblastoma (GBM), are the most common types of brain tumors with very poor prognosis. Systemic administration of various agents for the treatment of GBM has low efficiency due to the challenges of the drugs to cross the blood–brain barrier (BBB) to reach the tumor and its microenvironment.

Mesenchymal stem cells (MSCs) are an attractive solution for GBM therapy as they can augment the drug delivery across the BBB into the heterogenous glioma site [3]. Previously, Ebudi et al. documented the ability of stem cells to selectively home into inflammatory regions and tumors which indicates the possibility to use stem cells as a targeting vehicle for drug delivery [4]. Tumor tropism of stem cells of various origin (i.e., hematopoietic, embryonic, and, most widely used, mesenchymal) was further proven in the preclinical glioblastoma models [5]. Taking into consideration that currently MSCs (derived either from bone marrow or umbilical cord blood, and adipose tissue) are used in several preclinical and clinical trials, MSCs-based drug-delivery platforms could be considered as an attractive novel approach for tumor theranostics [6].

The question of whether the glioma tropism of MSCs is an adaptation of the organism to resist tumor progression, or the adaptation of a tumor for further growth is yet not solved. As shown previously, stem cell migration occurs during various physiological processes, including organ development, cell turnover, wound healing, inflammation, and cancer progression [7]. In many aspects, a tumor can be viewed as a non-healing wound [8], to which the body reacts with inflammation and tissue reorganization similar to the response of an organism to injury. Therefore, MSCs’ involvement into regenerative processes can explain the migratory capacity of stem cells into the tumor site [8]. However, the reported therapeutic effect of MSCs requires further exploration. Novel data indicate that the paracrine cell regulation of MSCs is enabled by the secretion of cytokines, soluble factors, extracellular vesicles, various proteins, microRNAs, mitochondria, etc. [9]. Indeed, several studies demonstrated that MSCs exhibit an antineoplastic effect via their secreted factors or they directly interact with cancer cells and other components of the tumor microenvironment [10,11]. Of note are other novel therapeutic and diagnostic (theranostic) technologies including nanoparticles (NPs) that are currently employed as a monotherapy or in combination with MSCs. However, NP-based drug-delivery systems (DDSs) still have many limitations, such as poor oral bioavailability, unstable circulation, inadequate distribution in tissues, and plausible toxicity [12]. Most of these disadvantages can be overcome by using MSCs as a carrier to load NP-based DDS due to the shielding effect of stem cells [13,14].

The in vivo analysis of biodistribution, bioavailability, and intratumoral accumulation of MSCs still remains a challenge due to the inaccuracy of the applied techniques and/or the influence of the tagged molecule on the physiological stem cell behavior. Currently, several approaches were employed for the evaluation of the biodistribution of MSCs, including PCR, labeling of MSCs with fluorescent lipophilic vital dyes or exogenously introduced markers, followed by a counting of the labeled cells in a region of interest (ROI) [15]. In contrast to these approaches, the whole-body imaging techniques (e.g., bioluminescent imaging, magnetic resonance imaging (MRI), positron emission, or single photon emission tomography, etc.) provide highly sensitive and accurate spatial biodistribution analysis of the administered cells [16].

In the current study the biodistribution and tumor-targeting potential of MSCs was assessed in a preclinical orthotopic model of C6 glioblastoma in Wistar rats. MSCs were labeled with superparamagnetic iron oxide nanoparticles (SPIONs) that have been previously demonstrated as a highly sensitive MR negative contrast agent (due to their unique physico–chemical properties) with a low cytotoxicity profile [17,18,19,20]. The biodistribution of SPIONs-labeled MSCs was determined by a highly sensitive method of longitudinal nonlinear response (NLR-*M_2_*) to a weak ac magnetic field, in which the second harmonic of magnetization *M*_2_ in dependence on the steady field *H* parallel to the ac field was registered [17,21]. Herein, we report the glioma-tropism of the administered MSCs and the feasibility of the SPIONs delivery to the tumor site.

## 2. Materials and Methods

### 2.1. Isolation, In Vitro Culture

The study was carried out using primary MSCs extracted from abdominal aorta of Wistar rats as described in Li et al. [22]. The rat C6 glioblastoma cell line was obtained from the shared research facility “Vertebrate cell culture collection” of the Institute of Cytology of the Russian Academy of Sciences (Saint Petersburg, Russia) supported by the Ministry of Science and Higher Education of the Russian Federation (Agreement no. 075-15-2021-683). Cells were cultivated in DMEM/F12 medium (Gibco, Carlsbad, CA, USA) supplemented with 10% fetal bovine serum (FBS) (Termo Fisher Scientific, Carlsbad, CA, USA) and 50 μg/mL gentamicin (Gibco, Carlsbad, CA, USA) at 37 °C and 5% CO_2_. Primary cells obtained from the fourth or sixth passages were used for experiments. Before the experiments, cells were harvested in the log phase of growth, and their viability was determined by 0.4% trypan blue exclusion. The immunophenotype of the rat MSCs was verified with a flow cytometer CytoFLEX S (Beckman Coulter, Indianopolis, Indiana, USA) using CD90 and CD45 cell markers as previously described [23]. Additionally, MSCs obtained from passage four were tested for the differentiation potential into osteoblasts, adipocytes, and chondroblasts in vitro (Appendix A).

### 2.2. In Vitro Rat MSCs Labeling with SPIONs

SPIONs were prepared as previously described [17]. The rat MSCs that reached a monolayer were incubated overnight with nanoparticles at a concentration of 150 μg/mL in a CO_2_ incubator. MSCs without SPIONs were used as a negative control. Cell viability was assessed by staining with a 0.4% Trypan blue solution (Biolot, Saint Petersburg, Russia). The cytotoxicity of nanoparticles was analyzed using the Vybrant^®^ MTT kit in accordance to the manufacturer’s protocol (Life Technologies, Carlsbad, CA, USA). To demonstrate ferric iron incorporation into cells, the Prussian Blue Staining of Cells in Culture was used in accordance to the manufacturer’s protocol (BioPAL, Worcester, MA, USA). Briefly, following 24 h of co-incubation with SPIONs, the culture medium was removed, cells were washed with phosphate-buffered saline (PBS), fixed with 4% paraformaldehyde, and mounted into the fluorescent mounting medium supplemented with 4′,6-diamidino-2-phenylindole (DAPI) (Abcam, Cambridge, GB). For the analysis of SPIONs intracellular localization and detection of iron labeled MSCs in brain tumor, the reflective laser (504 nm) scanning was applied. Cells were analyzed using a confocal microscope (Olympus FV3000) with an Olympus IX83 microscope confocal system (Olympus Corp., Tokyo, Japan). Additionally, for transmission electron microscopy (TEM) analysis of the SPIONs intracellular localization, cells following dehydration in an ascending series of ethanol, the specimens were embedded in a low viscosity “Spurr” embedding medium (Ted Pella, Redding, CA, USA). Ultrathin sections were cut with a Reichert Jung ultracut E microtome, mounted on nickel grids, contrasted with uranyl acetate and lead citrate, and examined employing Libra 120 electron microscope at 80 kV (Zeiss, Gina, Germany).

### 2.3. In Vitro Assessment of Rat MSCs Glioma Tropism

#### 2.3.1. Impedance-Based xCelligence System

The real-time rat MSCs migration assay was performed using xCelligence (Agilent, Santa Clara, CA, USA). To assess cell migration, 1 × 10^4^ rat MSCs were plated in serum-free media (upper chamber) and allowed to migrate toward the lower chamber containing a chemoattractant (C6 cells, conditioned medium from C6 cells harvested after 24 h culture with 5% FBS). The lower chamber contained serum-free medium which was used as a control. The cell index was acquired every 10 min for a period of 12 h. Four independent measurements were performed. For calculation of the kinetics of cell migration, automated analysis xCelligence software (Agilent, Santa Clara, CA, USA) was employed, according to the standard protocol.

#### 2.3.2. High-Content Quantitative Image Cytometer CQ1

The real-time rat MSCs migration assay was performed using a high-content quantitative image cytometer CQ1 (Yokogawa, Tokyo, Japan) with the Nipkow spinning disk confocal technology. Preliminarily C6 cells and rat MSCs were stained by PKH67 Green and PKH26 Red Fluorescent Cell Linker Kit for General Cell Membrane Labeling, respectively, in accordance to the manufacturer’s protocol (Sigma-Aldrich, St. Louis, MO, USA). Briefly, C6 cells were seeded in a 35-mm µ-Dish (IBIDI, Gräfelfing, Germany) and incubated until the cell confluence reached 90%. Then, the culture medium was removed, and a two-well culture insert (removable silicon gasket with two 70-μL wells) was placed in the center of the dish. Migration assays were performed by seeding MSCs into the two-well culture insert. Following cell attachment, the MSCs migration was visualized using time-lapse imaging. A fresh DMEM/F12 medium supplemented with 5% FBS and a conditioned medium from C6 cells harvested for 24 h culture was applied. Images (resolution of 2560 × 2160 pixels with a pixel size equivalent to 0.33 μm in x and y axis) were obtained using 488 nm, 561 nm lasers, phase contrast illumination and 10× objective lens every hour for a period of 12 h. For calculation of the migration area, the automated image analysis CQ1 software was employed according to the standard protocol with modifications. The rate of MSC migration was determined by analyzing the slope of the line within 0 and 12 h interval.

### 2.4. Animals

Male Wistar rats weighing 280–320 g were obtained from an animal nursery Rappolovo (Saint Petersburg, Russia). All animal studies were approved by the local ethical authorities of the Institute of Cytology (Saint Petersburg, Russia) and were in accordance with institutional guidelines for the welfare of animals.

### 2.5. Model of Intracranial C6 Glioblastoma and Administration of the SPION-Labeled MSCs

Before the operation the animals were anesthetized with intraperitoneal injection of Zoletil-100 (Virbac, Carros, France) and 2% Rometar (Bioveta, Ivanovice na Hané, Czech Republic) solution. The GFP-labeled C6 cells were used to induce glioblastoma in rats as described previously [24]. Cells were harvested in a log phase of growth and stereotactically infused into the *nucl. caudatus dexter* of anesthetized adult Wistar rats (0.5 × 10^6^ cells). On the 15th day following intracranial implantation of C6 cells animals were divided into two groups (six animals each): (1) the experimental group where SPION-labeled rat MSCs were intravenously administered via the tail vein (1.5 × 10^5^ cells; 200 μL); and (2) the control group injected with a PBS solution (200 μL).

### 2.6. Magnetic Resonance Imaging

The accumulation of the nanoparticle-labeled MSCs in the glioma site was evaluated employing 7.1 T high-field Bruker Avance 400 NMR spectrometer (Bruker–Biospin, Germany) using the following regimes: RARE-T1, TurboRARE-T2, and FLASH (gradient echo).

### 2.7. Histological Analysis

At the designated time point (24 h), animals were euthanized by CO_2_ asphyxiation, and unfixed samples of brain were stained with the H and E (Hematoxylin and eosin stain) method [25] and analyzed by transmission microscopy. For the detection of iron labeled MSCs in brain tumor, the extracted samples were fixed in 10% formalin, embedded in Tissue-Tek^®^ (Sakura Finetek Europe BV, Alphen an den Rijn, The Netherlands) and stored at −80 °C. Samples were sectioned using cryostat (Leica CM1850, Nussloch, Germany) and mounted in the mounting medium with DAPI (Abcam, Cambridge, UK). Fluorescence microscopy (EVOS Imaging System (Life Technologies, Carlsbad, CA, USA) was used for the analysis of frontal sections of the brain (16 μm thick).

### 2.8. In Vivo Analysis of the SPIONs-Labeled MSCs by NLR-M_2_

The extracted organs, i.e., brain (normal tissue and tumor), kidney, pancreas, lung, liver, spleen, heart, muscle, and skin were fixed in 10% formalin. To assess the biodistribution of the SPION-loaded rat MSCs in organs and tissues of animals, the registration of the second harmonic of magnetization *M_2_* generated in the material under the steady magnetic field *H* and the parallel weak ac field *h*·sin(*2πft*) with *h* = 13.8 Oe and *f* = 15.7 MHz was applied. Both signal components, Re*M_2_*(*H,T*) and Im*M_2_*(*H,T*), were recorded simultaneously as functions of *H* at various temperatures *T*. The steady field *H* was slowly scanned in the range from −300 to 300 Oe and backwards symmetrically relative to the point *H* = 0 with the cycle frequencies *F*_sc_ = 0.25 and 8 Hz to control the field hysteresis in the signal, which appearance indicates the presence of a spontaneous ferromagnetic moment in the sample.

To quantify the data of the NLR*-M_2_* measurements, the obtained dependences Re*M_2_*(*H,T*) and Im*M_2_*(*H,T*) were processed with the formalism based on the numerical solution of the kinetic Fokker–Planck equation for superparamagnetic particles, as it has been done previously in the study of the biodistribution of mesenchymal stem cells administrated in rabbits [17]. The output of the formalism is the function describing the magnetic field dependence of the real and imaginary parts of the nonlinear response to be applied for fitting the signal. The criterion of applicability of the procedure was the absence or only a small magnetic hysteresis of the *H* dependences of the signals which points on the superparamagnetic nature of the response. The computational resources of PIK Data Processing Centre of NRC “Kurchatov Institute”—PNPI (Gatchina, Russia)—were used, with the in-house provided software. Prior to the fitting, the raw data were averaged between the direct and reverse scans and antisymmetrized relative to *H* = 0 as required by the model. A set of parameters largely characterizing the SPION ensembles in the rats’ organs under study was obtained, describing their magnetic properties such as the distribution of the values of magnetic moments, magnetic anisotropy and magnetization dynamics. The most relevant parameter for the assessment of the biodistribution is the number of magnetically active centers in each organ derived from the saturation magnetization and the mean magnetic moment.

### 2.9. Statistical Analysis

For the analysis of two continuous variables, the parametric Student’s *t*-test was used. The significance level was equal to alpha = 0.05 for all tests with the confidence intervals at the 95% level. For comparison of multiple groups which had few observations, a nonparametric analog to the one-way ANOVA test, the Kruskal-Wallis test, was used.

## 3. Results

### 3.1. Detection of Cellular Internalization of Nanoparticles

Rat mesenchymal stem cells (MSCs) isolated from the rat aorta constituted a homogeneous population with a bipolar form (Figure 1A). The incorporation of SPIONs with a dimension of less than 50 nm and a surface charge of −12.7 mV (Figure 1B) by cells at the ultrastructural level was analyzed by TEM. The cytoplasm contained prominent heteromorphous endosomes with an average diameter of 0.5 μm filled with aggregates of numerous small electron dense particles (Figure 1C). MTT analysis did not reveal any toxic effect of the nanoparticles on the viability of MSCs at a concentration of 150 μg/mL (Figure 1D). Following co-incubation of MSCs with SPIONs, the Prussian blue staining and the subsequent reflective laser scanning by confocal microscopy demonstrated inclusion of nanoparticles into the cytoplasm (Figure 1E,F). The internalization of SPIONs in MSCs was confirmed by magnetic NLR-*M_2_* measurements of the colloidal solution of MSCs after co-incubation with SPIONs (see below).

### 3.2. Assessment of Rat MSCs Tropism to Glioma C6 Cells In Vitro

The results of the migration test performed on xCelligence system show the increased of the migration activity of MSCs in response to the presence of C6 cells (red curve) and the conditioned medium from C6 cells (green curve) in the lower chamber compared to the control media (blue and magenta curves) (Figure 2A). Evaluation of the migration of MSCs (red color) towards C6 cells (green color) was carried out using a high-content quantitative image cytometer CQ1. The stained MSCs were selected and their area was calculated over a period of 0–12 h (Figure 2B). It was shown that the presence of C6 cells and/or the conditioned medium from C6 cells can increase the rate of MSCs migration (Figure 2C,D, respectively).

### 3.3. Detection of SPION-Labeled Rat MSCs

GFP-C6 glioma cells were distinguished histologically against the background of normal brain tissue as a basophilic-colored area on the H and E staining during the transmission microscopy (Figure 3A) or as a GFP-positive area during the fluorescence microscopy (Figure 3B). Following 24 h of intravenous injection SPION-labeled rat MSCs were detected as homogenously distributed throughout the glioma tissue (Figure 3C). MR imaging of MSCs retention in the brain tumor with the help of high-field scanner yields the heterogeneous distribution of SPION-labeled stem cells in the tissue. The evaluation of the obtained MR sequences allowed us to assume that the ‘dark’ areas detected in the glioblastoma represent the accumulation of iron oxide nanoparticles that resulted, additionally, in the drop of T2 values (data not shown). (Figure 3D).

### 3.4. Biodistribution Analysis of SPIONs-Labeled Rat MSCs by NLR-M_2_

A full scope of the *NLR-M_2_* measurement data for the two representative rats (i.e., with incorporated SPIONs one day after injection and the control one) is shown in Figure 4 (tumor, brain, lungs, spleen, liver, and muscle) and Appendix A (kidneys, heart, pancreas, and skin). The real and imaginary parts of the second harmonic response as functions of the steady field are presented all normalized on the masses of the probes. The signals from the injected rats shown in Figure 4 demonstrate a weak field hysteresis that decreases with the decrease in the scanning frequency of the steady magnetic field, which is characteristic of nanoparticles in the superparamagnetic mode. This enabled the signals to be processed with the formalism described above. The hysteretic signals from the control animals indicate the presence of biogenic nanoparticles inside the organs. Unlike the data of Figure 4, the data in Appendix A are not processed as the control signals from these organs are comparable to or even exceed the signals from the tissues with the foreign SPIONs except of skin in which an inverse ratio was observed. But the skin response of the injected rats exhibits large, practically the same field hysteresis at both scan frequencies. This suggests that the magnetic fraction has not preserved superparamagnetic features and thus the processing formalism is inapplicable.

The results of the data treatment are shown in Figure 5. The solid curves are best fits of the signals for the organs of one or another rat presented in Figure 4.

In the solution prepared for injection, SPIONs form a highly dispersed system of aggregates with a mean content of 160 nanoparticles per aggregate and with the stem cells containing on average 8.27 × 10^4^ aggregates each [17]. As shown before [21], SPIONs in an aggregate are strongly magnetically coupled and therefore, the aggregates represent magnetically active centers rather than singular SPIONs. As the aggregates are magnetically uncoupled, the response signal is proportional to the number of aggregates within a sample. By comparing the signals from the organs to that of the initial solution with a known content of stem cells, the number of stem cells in the organs can be estimated. The *M*_2_ response from SPION-labeled MSCs in the PBS colloidal solution and its best fit are displayed in Appendix A and the obtained parameters are presented in Table 1 where
$\tilde{M}$ is the saturation magnetization, *N_P_* is the number of magnetically active centers, *N_MSC_* is the number of stem cells, *M_C_* is the mean magnetic moment, *σ* is the magnetic moment distribution width, *α* and τ*_N_* are the damping factor and the longitudinal free-relaxation time, respectively, characterizing the magnetization dynamics, and *E_A_* is the magnetic anisotropy energy.

In Table 1 the full scopes of the fitting parameters are presented only for the processed signals of the experimental or control animals. The values
$\tilde{M}$, *N_P_*, and *N_MSC_* for the same organ of the other rat, which signal was not processed, are obtained from correction of the corresponding *M*_2_ response amplitudes by the mass ratios of the similar organs assuming close values of the moments of their magnetic centers. The signals from the pancreas and skin of both rats were not processed and the estimations of
$\tilde{M}$ were obtained by the comparison of their *M*_2_ response amplitudes and the organ masses with those of the brain for each rat, based on the similarity of their signals. The data presented in Table 1 on
$\tilde{M}$ and the amounts of SPION-labeled MSCs in different tissues give one the information on biodistribution of MSCs in the animals. The visualization of this information is shown in Figure 6.

The values *M_C_*, σ, α, *τ_N_*, and *E_A_* in Table 1 are close to these of the initial solution [17] indicating that the aggregates have not changed since the injection and so can be used for the assessment of *N_MSC_* in the organs. In the order of increase of *N_MSC_*, the organs can be arranged as follows: pancreas, brain, skin, spleen, lungs, tumor, muscle, and liver for the first animal and brain, pancreas, muscle, spleen, tumor, skin, liver, and lungs for the second one.

## 4. Discussion

Currently, stem cell-based therapies are one of the most important emerging medical fields indifferent pathophysiological conditions and diseases, such as cardio-vascular diseases [26,27,28], autoimmune diseases [29,30], hereditary genetic conditions [31], and cancer [32,33]. Mesenchymal stem cells have attracted an increased attention as cellular vehicles for cancer therapy due to their low immunogenicity, fast ex vivo expansion, and inherent tumor-tropism and migratory properties. Despite the fact that a number of studies have shown that there is no risk of MSCs-induced tumorigenesis, there is still concern about stem cell-induced factors that might stimulate the growth of tumor cells [34]. The effects of MSCs on different types of tumor cells are still not completely assessed [33]. However, the ability of MSCs to across the blood–brain (BB) barrier could be used for glioma targeting and further development of stem cell-based therapies in neuro-oncology [20,35,36,37,38,39].

One of the most important mechanisms mediated by MSCs, is their strong tropism to the sites of injury and/or inflammation. Indeed, numerous chemokines, and growth factors can improve the targeting properties of MSCs towards gliomas [40]. Furthermore, as shown recently MSCs can interact with host immune cells and thereby regulate inflammatory responses [41]. Additionally, MSCs can induce indirect effects through secreted cytokines that significantly inhibit the proliferation of C6 cells, but also facilitate tumor cell migration and invasion [42]. However, the mechanisms underlying the tropism of MSCs to gliomas are not completely understood. It was shown that glioma cells during the process of proliferation (but not in resting and differentiated status) display a strong tropism towards MSCs and express certain chemokine factors that mediate cell migration [43].

In the current study, we have analyzed the tropism of MSCs to C6 glioma cells by two different methods. Indeed, in vitro analysis revealed that both C6 cell and conditioned medium from the supernatant can induce migration of MSCs (Figure 2). Subsequent in vivo studies confirmed that MSCs exhibit the tropism towards rat C6 glioblastoma cells. Thus, when administered intravenously cells were detected inside the tumors 24 h following injection (Figure 3). One of the limitations of the study is the absence of the analysis of the delivered MSCs’ survival and engraftment in the glioma tissues. Taking into account that under certain conditions (e.g., TGFß) in tumor microenvironment MSCs can differentiate into cancer-associated fibroblasts (CAFs) that are capable of promoting tumorigenesis the fate of the administered stem cells should be addressed in subsequent studies. Apart from detection of SPION-labeled MSCs in the tumor, signals from control animals [44,45,46,47] (presented in Figure 4 and Appendix A) together with the signals from rats injected with nanoparticle-labeled MSCs indicate the presence of biogenic nanoparticles in many organs. This phenomenon was observed by our group earlier in different animal organs [14,18,19] as well as in viable eukaryotic cells [21]. The kidneys, heart, and pancreas signals of the control rats (presented in Appendix A) are comparable to or even exceed the signals from the tissues obtained from the experimental animal. Presumably, this could be explained either by an off-target accumulation of SPION-labeled MSCs or by their degradation with subsequent excretion in these organs. The large hysteretic *M*_2_ response from the skin of the MSCs injected animals reveals practically the same field hysteresis at both scan frequencies. This suggests that the state of the magnetic fraction is changed from the ensemble of the single domain SPIONs to an ensemble of multi-domain particles formed by magnetic nuclei of SPIONs, for example, due to the plausible aggregation of particles following the degradation of the dextran coating.

As shown in Figure 4, Figure 5 and Figure 6 and Table 1 the amplitudes of the signals strongly differ between the organs up to two orders of magnitude (brain and lungs of the second animal) and approximately by one order between the pancreas and the brain in comparison with the liver, tumor, and spleen of both animals indicating an inhomogeneous biodistribution of SPIONs (and MSC, respectively) in the animal. Responses of the same organ and MSC accumulation in different rats are similar in the liver, differs approximately 1.5 ÷ 2 fold in the brain, tumor, spleen, and pancreas and differs up to 4 ÷ 6 fold in the muscle, skin, and lungs. The qualitative similarity of MSCs’ biodistributions in the most organs of rats (see Figure 6) seems to signify a common feature while their difference in the muscle, skin, and lungs evidences some role of the animal individuality on MSC retention. The lungs accumulated the largest amount of SPIONs whereas the brain and pancreas amass the lowest amount. This biodistribution pattern is in line with previously reported data on the biodistribution of MSCs [33]. Presumably, employing novel pharmacokinetic models for predicting the MSCs behavior in the organism and their accumulation in the tumor tissues could improve the bioavailability of the cells and estimate the exact amount of administered MSCs [48].

In the current study for targeting glioma, MSCs were intravenously administered. Future research direction to increase the local concentration of stem cells could be based on application of biodegradable materials (e.g., fibrin scaffolds, cryogel, etc.) that are directly implanted into the tumor site [49,50,51].

## 5. Conclusions

These results highlight the potential of MSCs as a novel tumor-targeting strategy for diagnostics and therapy of malignant brain tumors.

## Figures and Tables

**Figure 1 biomedicines-09-01592-f001:**
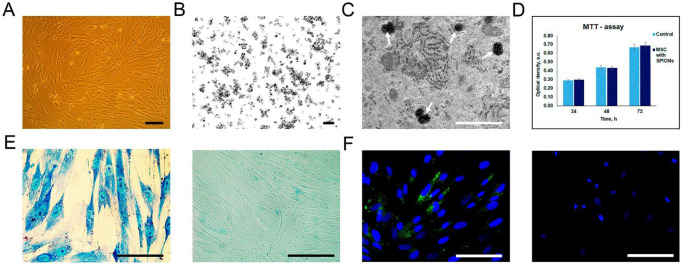
Assessment of nanoparticles internalization by rat mesenchymal stem cells (MSCs). (**A**) Live cell imaging. Scale bar, 100 μm. (**B**,**C**) TEM image of the SPIONs accumulated in the MSCs. White arrows point to secondary endosomes filled with heterogeneous material, including electron-dense nanoparticles. Scale bars: (**B**,**C**) 100 nm and 1 μm, respectively. (**D**) The influence of the SPIONs on the viability of cells, MTT data. (**E**) Prussian blue staining of the SPION-loaded MCSs and control (non-treated cells), respectively. (**F**) Confocal microscopy images of rat MSCs co-incubated with SPIONs and control (non-treated cells), respectively. Nuclei were stained with DAPI (blue). SPIONs were detected by reflective laser scanning (green). Scale bars: (**E**,**F**) 50 µm.

**Figure 2 biomedicines-09-01592-f002:**
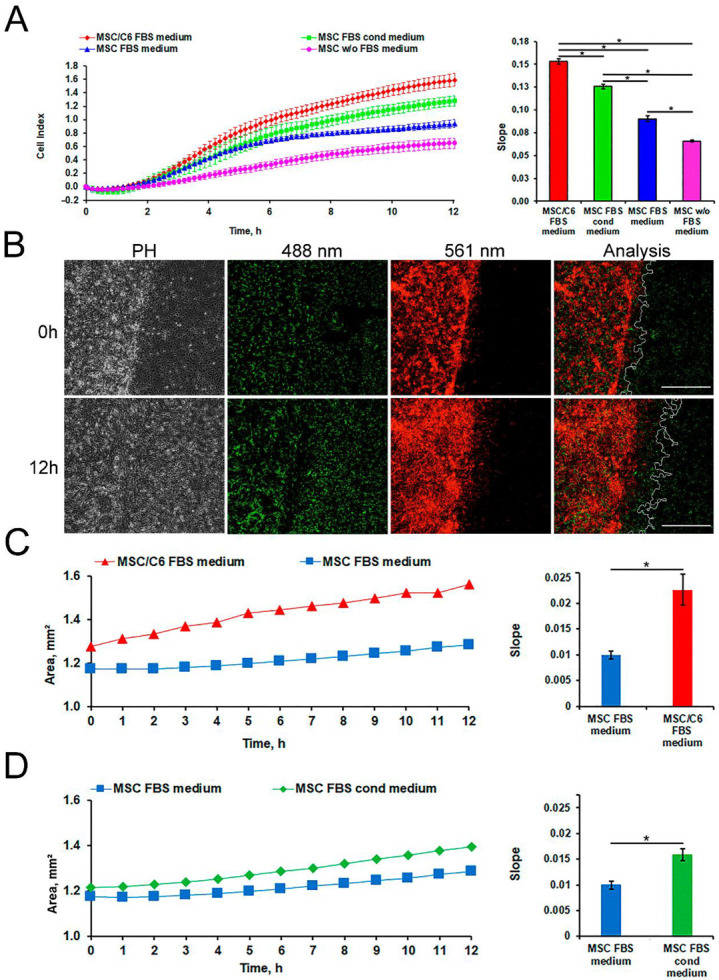
Evaluation of rat MSCs tropism to glioma C6 cell in vitro. (**A**) Imaging cell migration results with the xCelligence system. Levels of significance between all experimental and control group (serum-free medium) are shown as * *p* < 0.05. (**B**) Confocal microscopy images of MSCs migration using a cytometer CQ1. C6 cells (green color) and MSCs (red color) were detected using a diode laser (488 and 561 nm, respectively). Scale bars, 500 µm. The presence of C6 cells (**C**) and a conditioned medium from C6 cells (**D**) increased the rate of MSCs migration. Levels of significance between experimental and control group are shown as * *p* < 0.05.

**Figure 3 biomedicines-09-01592-f003:**
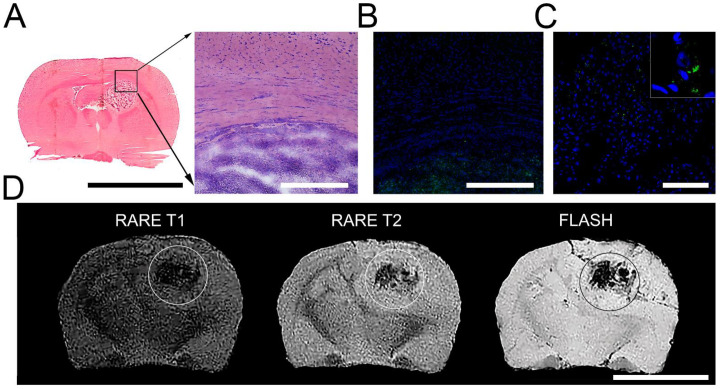
Evaluation of rat MSCs tropism to glioma in vivo. (**A**) Frozen section of animal brain with glioma (black square). Scale bar, 10 mm. Borders of glioma and brain tissue at high magnification. H and E staining. (**B**) Fluorescence microscopy images. Detection of the glioma area. Nuclei were stained with DAPI (blue) and GFP-positive C6 cells (green). Scale bars, 400 µm. (**C**) Confocal microscopy image. Cryosection of brain with glioma following 24 h after intravenous administration of the SPION-labeled MSCs. Scale bars, 100 µm. (**D**) Magnetic resonance images of the brains of glioma-bearing animals following 24 h after intravenous administration of the SPION-labeled MSCs. Images were obtained at RARE-T1, RARE-T2, and FLASH regimens. Scale bar, 10 mm.

**Figure 4 biomedicines-09-01592-f004:**
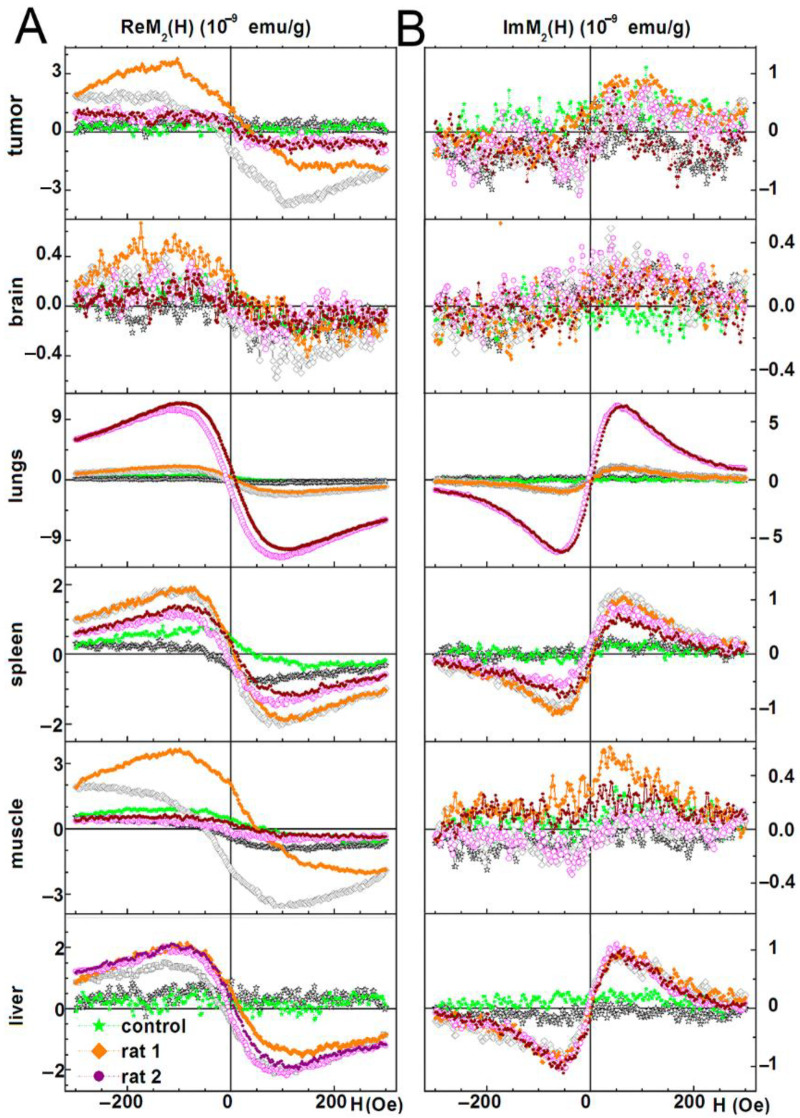
Real and imaginary parts of the nonlinear magnetic response as functions of the dc magnetic field direct (filled symbols) and reverse (open symbols) scans with *F*_sc_ = 8 Hz are presented for liver, tumor, brain, lungs, spleen, and muscle extracted from two rats at 24 h following injection of SPIONs-labeled MSCs and one control rat without injection. Panels (**A**,**B**) present real and imaginary parts of M_2_ response accordingly.

**Figure 5 biomedicines-09-01592-f005:**
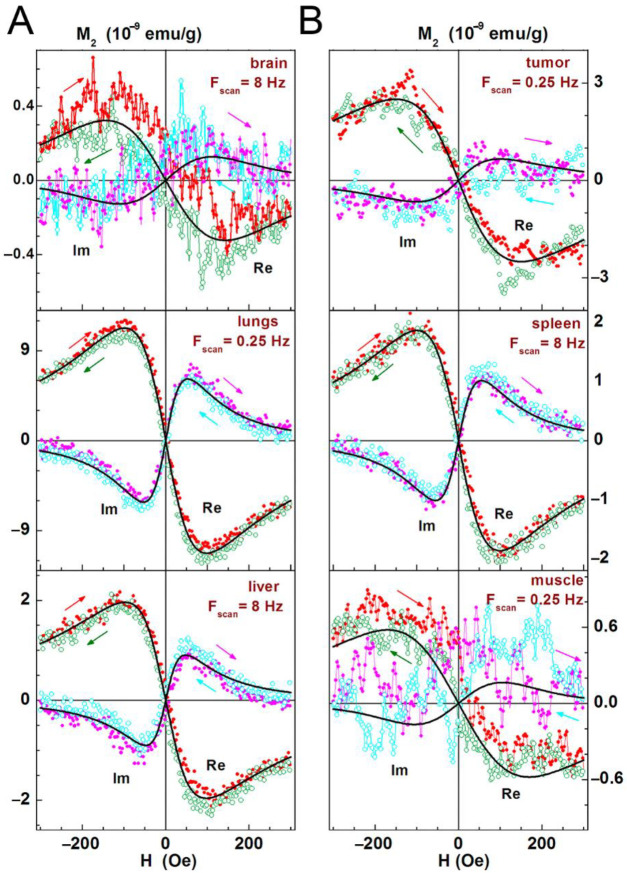
Real and imaginary parts of the nonlinear magnetic response as functions of the dc magnetic field direct (filled red and magenta symbols) and reverse (open olive and cyan symbols) scans (indicated additionally by arrows of the same with symbols colours) at *F*_sc_ = 8 or 0.25 Hz, with their best fits are displayed for brain, lungs, liver (panel **A**), and tumor, spleen, muscle (panel **B**) following one day after intravenous injection of SPION-labeled MSCs. Only every 10th experimental point is shown. Obtained parameters are presented in Table 1.

**Figure 6 biomedicines-09-01592-f006:**
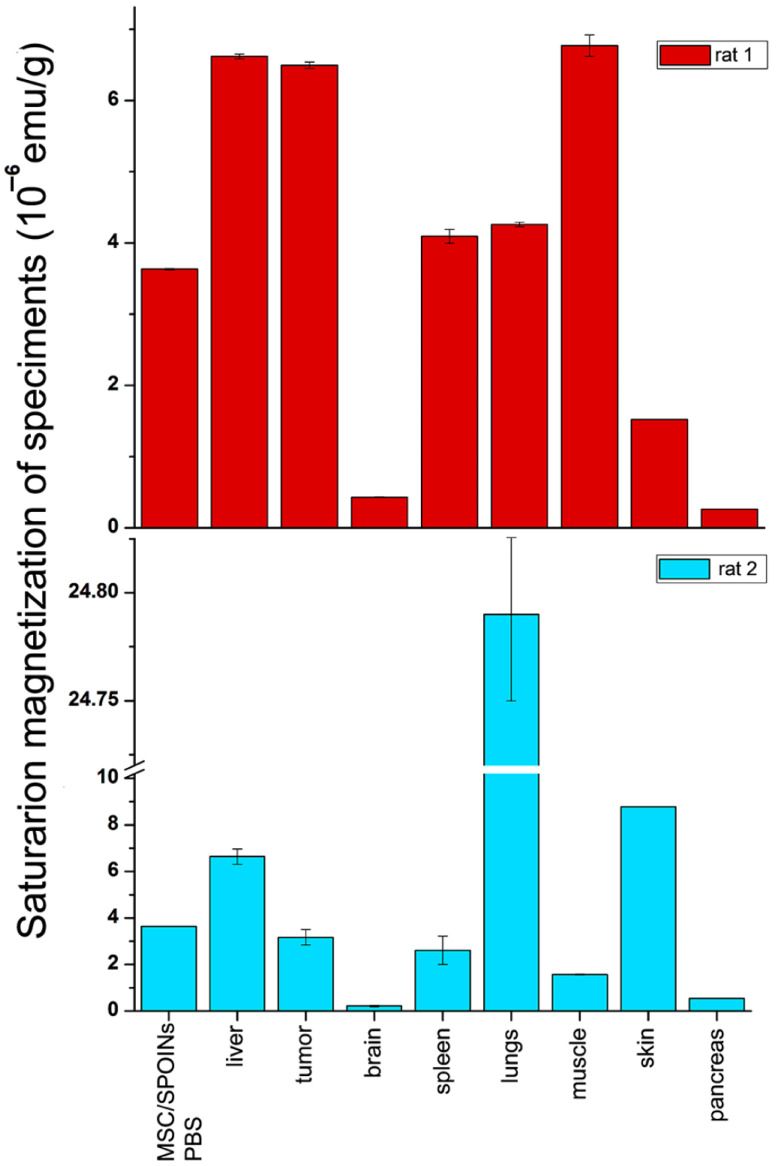
Representative diagrams of biodistribution of the SPIONs-labeled rat MSCs in tissues of experimental and control animals.

**Table 1 biomedicines-09-01592-t001:** Parameters of M2 signals and biodistribution of SPION-labeled MSCs in rats.

Organ	$\tilde{M}$, emu/g	*N_P_*, g^−1^	*N_MSC_*, g^−1^	*M_C_*, *μ_B_*	*σ*	*α*	*τ_N_*, ns	*E_A_*, K
MSC-SPIONs in PBS	3.633(5) × 10^−6^	1.24(4) × 10^10^	1.5 × 10^5^	31,580(30)	0.734(1)	0.2057(6)	1.020(3)	8.3(1.0)
Tumor (rat 1)	6.495(45) × 10^−6^	2.65(8) × 10^10^	3.2 × 10^5^	26,400(100)	0.485(5)	0.287 (9)	0.651(21)	0(15)
Brain (rat 1)	4.303(57) × 10^−7^	1.18(2) × 10^9^	1.4 × 10^4^	39,400(300)	0.241(17)	0.207 (8)	1.29(5)	0(29)
Lungs (rat 1)	4.26(3) × 10^−6^	1.74(5) × 10^10^	2.1 × 10^5^	-	-	-	-	-
Spleen (rat 1)	4.093(96) × 10^−6^	1.578(51) × 10^10^	1.9 × 10^5^	28,000(600)	0.758(8)	0.2374(13)	0.810(18)	17.4(9)
Muscle (rat 1)	6.77(15) × 10^−6^	2.85(47) × 10^10^	3.4 × 10^5^	-	-	-	-	-
Liver (rat 1)	6.618(33) × 10^−6^	3.24(24) × 10^10^	3.9 × 10^5^	-	-	-	-	-
Skin (rat 1)	~1.52 × 10^−6^	~4.17 × 10^9^	~5.0 × 10^4^	-	-	-	-	-
Pancreas (rat 1)	~2.64 × 10^−7^	~0.72(9) × 10^9^	~8.7 × 10^3^	-	-	-	-	-
Tumor (rat 2)	3.17(33) × 10^−6^	1.29(20) × 10^10^	1.6 × 10^5^	-	-	-	-	-
Brain (rat 2)	2.17(3) × 10^−7^	0.59(2) × 10^9^	7.1 × 10^3^	-	-	-	-	-
Lungs (rat 2)	2.479(57) × 10^−5^	1.01(3) × 10^11^	1.2 × 10^6^	26,700(600)	0.786(8)	0.2421(12)	0.759(17)	13.3(1.2)
Spleen (rat 2)	2.61(61) × 10^−6^	1.01(32) × 10^10^	1.2 × 10^5^	-	-	-	-	-
Muscle (rat 2)	1.572(23) × 10^−6^	0.662(11) × 10^10^	8.0 × 10^4^	25,700(0)	0.283(0)	1.036 (0)	0.332(0)	8.22(0)
Liver (rat 2)	6.64(33) × 10^−6^	3.48(24) × 10^10^	4.2 × 10^5^	20,600(1000)	0.862(15)	0.291 (2)	0.502(24)	0(12)
Skin (rat 2)	~8.78 × 10^−6^	~2.41 × 10^10^	~2.9 × 10^5^	-	-	-	-	-
Pancreas (rat 2)	~5.44×10^−7^	~1.49 × 10^9^	~1.8 × 10^4^	-	-	-	-	-

## Data Availability

Data available upon request.

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
