# Peer review of "Targeting Brain Tumors with Mesenchymal Stem Cells in the Experimental Model of the Orthotopic Glioblastoma in Rats"

_biomedicines, 2021, doi:10.3390/biomedicines9111592_

Round 1

Reviewer 1 Report

The Authors investigated the glioma-tropisms of Mesenchymal stem cells (MSCs), by monitoring the migration and distribution of the superparamagnetic nanoparticle-labeled MSCs in an orthotopic model of C6 glioblastoma in Wistar rats. The paper is overall clear and interesting. However, there are specific points that should be clarified:

Abstract: 
Please report the acronym in full length when mentioned for the first time. 

Introduction section 

The first sentence is too long and confusing. Please, modify it for clarity.

Material and Methods

The immunotype of the rat MSCs was verified by using only two markers, one positive and one negative, CD90 and CD45 respectively. However, the MSC characterization is more complex and requires the simultaneous detection of multiple markers such as CD44, CD90, CD105, CD73 among the positive and CD34, CD45, and CD14 referring to the negative ones. Furthermore, further proof of the mesenchymal origin of the cells is given by their ability to differentiate in mesenchymal cell lines, including bone cells, adipocytes, or chondrocytes. Therefore, the detection of only two markers is insufficient to demonstrate the isolation of MSCs. 

It is not clear how the conditioned medium was produced. In line 138 it is reported that conditioned medium deriving from C6 glioma cells was harvested after 24h  culture. What was the percentage of serum used to produce the conditioned medium? Please specify this detail. Clarify also lines 138-140.

Results
Figure 1:
-    The resolution of the pictures is very scarce. Therefore, it is impossible to evaluate the correspondence between the results content and the figure. Please, improve it. 
-    In the MTT figure, the histogram of the control should be placed before those of the treatment condition. Furthermore, the statistical significance must be reported by comparing the treatment condition with respect to the control at the same incubation time. However, in this graph, the viability of MSCs with SPION was compared to the control at a different time point. Considering that, at each time point, the viability of MSCs with SPION increases concomitantly to the control, there is no significant increment of cell viability of MSCs with SPION compared to the control. Please, correct it removing the statistical significance. 
-    In addition, it is not appropriate to define MTT assay as a proliferation assay. This technique allows testing the metabolic activity of viable cells. Accordingly, replace the term proliferation with the viability or metabolic activity of viable cells. 
-    In the figure caption, line 246, please correct the scale of the image. 

The resolution of all the figures must be improved.  

It is an interesting study. However, it is possible to recognize some limitations. Specifically, the distribution of MSCs in organs and tissues different from the organ target, and, more importantly, despite an enrichment of MSCs in the tumor site, it was not proved the role of MSCs as a tumor-targeting strategy. Therefore, this aspect should be reported in the discussion. 

Author Response

  • Abstract: Please report the acronym in full length when mentioned for the first time.

ANSWER: The acronym (SPION) at first mention was reported as superparamagnetic iron oxide nanoparticle.

  • Introduction section. The first sentence is too long and confusing. Please, modify it for clarity.

ANSWER: The first sentence was corrected.

  • Material and Methods
  • The immunotype of the rat MSCs was verified by using only two markers, one positive and one negative, CD90 and CD45 respectively. However, the MSC characterization is more complex and requires the simultaneous detection of multiple markers such as CD44, CD90, CD105, CD73 among the positive and CD34, CD45, and CD14 referring to the negative ones. Furthermore, further proof of the mesenchymal origin of the cells is given by their ability to differentiate in mesenchymal cell lines, including bone cells, adipocytes, or chondrocytes. Therefore, the detection of only two markers is insufficient to demonstrate the isolation of MSCs.

ANSWER: Additionally, rat MSCs were tested for the differentiation potential (Figure S1).

  • It is not clear how the conditioned medium was produced. In line 138 it is reported that conditioned medium deriving from C6 glioma cells was harvested after 24h culture. What was the percentage of serum used to produce the conditioned medium? Please specify this detail. Clarify also lines 138-140.

ANSWER: С6 cells were cultured with the DMEM/F12 medium containing 5% FBS for 24 hours, then the medium was centrifuged to remove debris and added to the lower well. The lower chamber contained serum-free medium was used as a control.

4) Results.

  • The resolution of the pictures is very scarce. Therefore, it is impossible to evaluate the correspondence between the results content and the figure. Please, improve it.

ANSWER: The figures 1, 2, 3, 6 were corrected.

  • In the MTT figure, the histogram of the control should be placed before those of the treatment condition. Furthermore, the statistical significance must be reported by comparing the treatment condition with respect to the control at the same incubation time. However, in this graph, the viability of MSCs with SPION was compared to the control at a different time point. Considering that, at each time point, the viability of MSCs with SPION increases concomitantly to the control, there is no significant increment of cell viability of MSCs with SPION compared to the control. Please, correct it removing the statistical significance.

ANSWER: The MTT figure was corrected.

  • In addition, it is not appropriate to define MTT assay as a proliferation assay. This technique allows testing the metabolic activity of viable cells. Accordingly, replace the term proliferation with the viability or metabolic activity of viable cells.

ANSWER: MTT analysis did not reveal any toxic effect of the nanoparticles on the viability of MSCs at concentration of 150 μg/ml (248 line)

In the figure caption, line 246, please correct the scale of the image. The resolution of all the figures must be improved. 

ANSWER: The scale of the image was corrected and resolution of all the figures were improved. 

  • It is an interesting study. However, it is possible to recognize some limitations. Specifically, the distribution of MSCs in organs and tissues different from the organ target, and, more importantly, despite an enrichment of MSCs in the tumor site, it was not proved the role of MSCs as a tumor-targeting strategy. Therefore, this aspect should be reported in the discussion.

ANSWER: Limitation of the study was added to the Discussion section.

Reviewer 2 Report

Review on the manuscript titled “Targeting Brain Tumors with Mesenchymal Stem Cells in the Experimental Model of the Orthotopic Glioblastoma in Rats” by  Yudintceva N et al., submitted to Biomedicines

Manuscript ID: biomedicines-1426542

Dear Authors,

Glioblastoma is the most aggressive type of brain tumors, and the outcome of the treatment remains poor. Mesenchymal stem cells (MSCs) provide a promising option for cell-based therapy. The authors studied the tumor tropism and distribution of MSCs labeled with the superparamagnetic nanoparticles in the orthotopic model of C6 glioblastoma. The results showed that MSCs are highly migratory towards glioblastoma cells in vitro; the labeled MSCs accumulate in the tumor in vivo; and MSCs are retained in the glioblastoma by histological analysis. The authors concluded that MSCs can be used as a drug delivery platform for cancer therapy.

Please consider the following:

  1. A graphical abstract summarizing the manuscript is highly recommended.
  2. Page 6,7,8,12 Figure 1,2,3,6: Please present higher quality of the figure.
  3. Pages 5,6,7, The Sections 3.1,3.2: Please present numerical statistical data in open form in the table.
  4. Pages 12-14, Discussion, Conclusion: Please present the previous and present studies, weaknesses or limitation in the present study, potentials, the ultimate goal, research or knowledge needed to achieve, the future research direction, and the biggest challenge in this goal, among others.
  5. Pages 14-16, References: Please cite more references, preferably more than 50.

The manuscript contains six figures, one table and 40 references. The quality of the manuscript can be improved. The manuscript carries important value presenting a potential role of MSCs as a drug delivery platform for glioblastoma therapy. I recommend this manuscript for publication after minor revision.

Author Response

  • A graphical abstract summarizing the manuscript is highly recommended.

ANSWER: A graphical abstract was added.

  • Page 6,7,8,12 Figure 1,2,3,6:Please present higher quality of the figure.

ANSWER: The figures 1, 2, 3, 6 were corrected and resolution of all the figures were improved. 

  • Pages 5,6,7, The Sections 3.1,3.2:Please present numerical statistical data in open form in the table.

ANSWER: The numerical statistical data in open form in the table were presented as an attachment.

  • Pages 12-14, Discussion, Conclusion:Please present the previous and present studies, weaknesses or limitation in the present study, potentials, the ultimate goal, research or knowledge needed to achieve, the future research direction, and the biggest challenge in this goal, among others.

ANSWER: This was added to the Discussion Section.

  • Pages 14-16, References:Please cite more references, preferably more than 50.

ANSWER: This was corrected.
